# Adult Diel Locomotor Behaviour in the Agricultural Pest *Plutella xylostella* Reflects Temperature-Driven and Light-Repressed Regulation Rather than Coupling to Circadian Clock Gene Rhythms

**DOI:** 10.3390/insects16020182

**Published:** 2025-02-08

**Authors:** Connor J. Tyler, Shubhangi Mahajan, Lena Smith, Haruko Okamoto, Herman Wijnen

**Affiliations:** 1SPITFIRE NERC Doctoral Training Partnership, SoCoBio BBSRC Doctoral Training Partnership, School of Biological Sciences and Institute for Life Sciences, University of Southampton, Highfield Campus, Southampton SO17 1BJ, UK; ct13g15@southamptonalumni.ac.uk (C.J.T.); sm6n22@soton.ac.uk (S.M.); ls9g20@soton.ac.uk (L.S.); h.okamoto@sussex.ac.uk (H.O.); 2School of Life Sciences, University of Sussex, Brighton BN1 9QG, UK

**Keywords:** diamondback moth, behavioural rhythm, daily timekeeping, light response, temperature response

## Abstract

Insect pests such as the diamondback moth, which infests cabbages and related crops, inflict massive amounts of damage to agriculture that costs billions of USD annually. Since the adult diamondback moths spread by migrating, factors controlling their activity and movement are of interest. Daily and seasonal activity rhythms in many animals are regulated by innate daily timekeeping systems termed circadian clocks. In this study, we describe rhythms in the molecular elements of this timekeeping system in diamondback moths and we test if the circadian clock can be held responsible for daily rhythms in moth activity. We found that diamondback moth activity was suppressed by light and regulated by environmental temperature rather than the circadian clock. These insights are helpful in predicting the activity of this agricultural pest.

## 1. Introduction

Climate change and an exponentially growing human population are putting pressure on arable land and threatening global famines before the end of the century [1], making food security a priority issue that needs to be addressed [2]. A key factor in the challenge to maintain food production is the control of pest species, with the effects of pests being underestimated [3] and increasing in severity as the climate changes [4,5,6,7]. *P. xylostella* (diamondback moth) is a pest species that has taken hold globally, having achieved a cosmopolitan distribution with a seasonally expanding range that covers all but the coldest latitudes [8,9]. *P. xylostella* is a *Brassicaceae* specialist herbivore [10] causing worldwide losses estimated at USD ~5 billion annually [11], up from USD ~1 billion in the 1990s, making *P. xylostella* arguably the world’s most costly lepidopteran pest [8]. *P. xylostella* challenges farmers due to rapid proliferation in newly colonised fields and is able to produce 250+ eggs per female [12] and complete multiple life cycles per season [8,13]. *P. xylostella* also shows rapid pesticide resistance gain [14,15,16], leading to control failures with some regions facing > 90% crop losses [17,18]. Integrated Pest Management (IPM) strategies are being implemented to both diminish pesticide use and *P. xylostella’s* impact [19,20,21]; however, the uptake of these strategies has been limited, with the prohibitive complexity of treatments needed for success causing farmers to fall back onto aggressive pesticide spraying protocols [22,23,24]. The study of behavioural manipulation as part of control strategies has been gaining increasing attention as these research pathways may complement and help increase the efficacy of IPM [25,26,27].

The circadian clock acts as an endogenous timekeeping mechanism controlling processes in organisms to best fit the daily rhythms of the planet [28,29]. This clock runs via a series of transcriptional and translational feedback loops (TTFL) that maintain ~24 h rhythms without the need for external stimuli such as light and temperature though it can be entrained by them. The lepidopteran circadian clock differs from that of the model organism *Drosophila melanogaster* [30,31], having two CRYPTOCHROME (CRY) proteins active in the TTFL. CRY2 (vertebrate-like CRY, absent in *Drosophila*) acts as negative regulator in the complex that represses *per, tim* and *cry2* expression [32,33].

Circadian rhythms have a widespread impact on gene expression. For example, ~40% of coding genes were found to exhibit circadian rhythms in mice [34] and ~35% in plants [35]. Synchrony of internal and environmental rhythms is important for health and well-being with circadian clocks exerting a strong influence over organisms’ immune systems and anticipation of threats [36,37,38,39]. Lepidopteran circadian mechanisms have been shown to influence numerous behaviours including life stage cycles, oviposition [40] and seasonal migrations [41]. *P. xylostella* may better adapt to local environments and rhythmic changes via seasonal migration. In other species, seasonal migration is known to be informed by the clock-dependent detection of relevant changes in the photoperiod [42,43] and temperature [44,45]. Moreover, the way in which *P. xylostella* controls and modulates its behaviour when invading new regions and after invasions have taken place may be important for informing control measures as well as complimenting and improving the efficacy of IPM strategies. *P. xylostella,* which lacks the ability to diapause, migrates seasonally, invading brassica crops grown at higher latitudes as temperatures warm with populations sustained as long as temperatures remain elevated [9,46,47]. The modulating photoperiod in a neotropical *P. xylostella* population had no discernible effect morphology or life history [48]. However, systematic analyses of the impact of changes in both the photoperiod and daily temperature cycles on *P. xylostella* physiology and behaviour remain to be conducted.

It is therefore important to understand both *P. xylostella*’s daily timekeeping mechanisms and the rhythmic behavioural outputs that occur in association with these. Molecular circadian rhythms underlying daily timekeeping were explored by determining transcript profiles of the *P. xylostella* clock genes *per* and *tim*. In addition, *P. xylostella* was studied for its diel locomotor activity under a range of environmental light and temperature cycles. Our behavioural analyses of the ROTH *P. xylostella* strain maintained on *Brassica rapa* complement a prior behavioural study using the Geneva 88 *P. xylostella* strain maintained on an artificial diet that identified nocturnal behaviour with relatively weak male-specific circadian behaviour [49].

## 2. Materials and Methods

### 2.1. P. xylostella Culture Maintenance

*P. xylostella* populations were reared within an environmental control room (ECR) at 20 °C in a 12/12 h Light/Dark (L/D) light cycle. W30 × D30 × H30 cm BugDorms (Watkins & Doncaster, Leominster, UK) were used to house mixed-sex populations. The population was acquired from Rothamsted Research, (ROTH strain wild collections during the 1960s) and has been continuously maintained on *Brassica rapa*. A sugar-water source was provided for adults during the egg-laying period, with adults mixed between new cages to prevent genetic isolation.

### 2.2. P. xylostella Locomotor Activity Assays

In preliminary activity monitor experiments, three different substrates were trialled. Cotton wool saturated with sugar-water or diluted honey versus sugar-agar solid media. As moths often got stuck and/or wet when sugar/honey solution-saturated cotton wool was used, the solid sugar-agar media were used instead. To control fungal growth, this media was prepared with 0.07% methyl 4-hydroxybenzoate. This was added using a 10% methyl 4-hydroxybenzoate w/v stock solution in ethanol after 1% agar and 5% table sugar media had been boiled and cooled down to ~60 °C. During preparation, 25 mm diameter × 95 mm length glass test tubes were filled from the bottom with 2 cm of the media. The media was left to dry before adding individual *P. xylostella* under CO_2_ anaesthesia, and the tubes were capped with cellulose acetate to allow gas exchange. Individual *P. xylostella* activity counts were detected with the tubes in horizontal orientation by the TriKinetics LAM25 assay system (TriKinetics, Waltham, MA, USA). Based on the increased mortality at 23 °C in preliminary experiments, assays were conducted at 17 °C or 20 °C.

For individual locomotor assays at a constant temperature, LAM25 monitors were kept inside a black plastic box at 17 °C, with local humidity provided by a tray of water containing algicide and biocide to suppress microbial growth. White LEDs were mounted inside the black plastic tub providing 8 µmol/m^2^/s (~400 lux) light for 12/12, 6/6/6/6, 14/10, 16/8, 18/6 and 20/4 h of light (L) and dark (D) cycles or in constant light (L/L) or constant darkness (D/D), shown in Table 1. In parallel, environmental cycles mimicking light and temperature recordings in Kent (51.2874° N, 0.4400° E), UK, in April and June, were set up for individual locomotor activity assays in an Percival incubator, model number: DR-36VL, (Clf Plantclimatics, Wertingen, Germany) based on environmental data from [50], producing the conditions shown in Table 1. The recorded environmental profile in the incubator setups is shown in Appendix A, where a programmable LED lamp (FLUVAL14521, Rolf C. Hagen (UK) Ltd. Castleford, UK) was used to create ramping dawn and dusk light, with the incubator maintaining the temperature profile and max light producing 38 µmol/m^2^/s (~2000 lux). Gradually ramping temperature cycles in (D/D) and (L/L) conditions were produced in the same incubators, using the four built-in fluorescent lights to provide the light for the latter condition. For each locomotor assay, *P. xylostella* was selected freshly from culturing enclosures and was recorded over 6+ days. The data were collected by a DAM (Drosophila Activity Monitor) system (Trikinetics, Waltham, MA, USA) [51] in 5 min segments and were subsequently exported to ClockLab Analysis Version 6 (Actimetrics, Lafayette, IN, USA) for analysis. DEnM monitors were used to keep track of experimental light, temperature and humidity conditions.

### 2.3. P. xylostella RNA Collection and Extraction

Whole adult *P. xylostella* male moths were maintained at 20 °C under 12 h:12 h L/D conditions and then collected every 4 h during the last day of L/D and the first day of subsequent D/D conditions (ZT1-5-9-13-17-21-CT1-5-9-13-17-21). Male moths were used exclusively to avoid confounding effects due to sex ratio fluctuations or female egg load [52]. Adult males were snap-frozen at −80 °C for storage and used to extract RNA using the RNAqueous-4PCR kit (Thermo Fisher Scientific, Loughborough, UK). The isolated RNA was treated with DNase 1 and the resulting RNA samples were tested for concentration and quality using a NanoDrop 1000 spectrophotometer (Thermo Fisher Scientific, Loughborough, UK). Samples were aliquoted into an appropriate working volume (used for specific quantitative polymerase chain reaction (qPCR) plate setups) and were kept at −20 °C for short-term storage.

### 2.4. qPCR Protocol

RNA samples were used as templates for one-step amplification reverse transcriptase quantitative PCR (qPCR) with the PrecisionPLUS OneStep qRT-PCR Master Mix (Primer Design Ltd., Chandler’s Ford, UK), with both a SYBR green florescent dye and ROX dye. qPCR was performed using a StepOnePlus Real-Time PCR System (Thermo Fisher Scientific, Loughborough, UK) to quantify proportional levels of specific RNA transcripts between samples [53]. Amplification thresholds were used to calculate the proportional level of primer-specific transcripts present in each RNA sample using the 2^−ΔΔCT^ method [54]. *Elongation factor 1 α* (*Ef1α*) was used as a housekeeping reference gene in order to analyse the relative expression of circadian genes *period* (*per*) and *timeless* (*tim*) shown in Table 2. Primers were designed using Primer-3 and Primer-BLAST [55,56] to amplify desired distinct gene transcripts. Adherence to the MIQE guidelines was submitted along with the research [57].

### 2.5. Statistical Analyses

ClockLab Analysis Version 6 software was used to analyse and produce graphical representations of *P. xylostella* locomotor activity data at a 30 min resolution in the form of double-plotted actograms, daily activity profiles and chi-square periodograms and to quantify specific associated parameters of both individual moths and populations. Individual moths were assumed to have died and were excluded from the results if their activity stopped permanently during a 6-day analysis interval. Flies were considered rhythmic if they exhibited a significant (*p* <= 0.01) chi-square periodogram amplitude for a period in the circadian range (15–35 h). Relative rhythmic power (RRP) was calculated as the ratio between that amplitude and the associated significance threshold value. Shapiro–Wilk tests were used to test if data met the assumptions required for parametric testing, e.g., normal distribution of data.

Where the assumption of normality was consistently found to be met, comparisons were performed using t-tests or ANOVA with Tukey’s multiple comparisons analysis. In cases where some of the data sets violated this assumption, nonparametric analyses (Mann–Whitney U test, Kruskal–Wallis tests with Dunn’s multiple comparison post hoc tests) were used instead. The differences between the light- and dark-phase activity counts of individual moths were assessed using Wilcoxon matched-pairs signed rank test. Simple linear regression was used to describe the relationship between dark-phase length and average dark-phase activity counts. The resulting equation was then compared to models assuming constant daily activity or a constant activity rate during the dark phase. Two-sided permutation t-test estimation statistics [59] were used to illustrate the distribution of activity across two halves of experimental days. The CircaCompare R package 0.1.1 [60] was used for the calculation and comparisons of rhythmic activity and expression profiles using rhythmic features (mesor, amplitude and phase) to compare rhythmic patterns between groups of data through cosinusoidal curve fitting.

## 3. Results

### 3.1. P. xylostella Circadian Clock Gene Rhythms

The rhythmic profiles of the clock gene transcripts for *tim* and *per* relative to a reference gene (*Ef1α*) were determined by qPCR from total RNA extracted from adult male moths across a 2-day L/D-D/D time course sampled at a 4 h resolution (Figure 1, Table 3). Significant rhythmicity was observed for each clock gene under each environmental condition using cosinusoidal fitting (Appendix A). The transcript profiles for *per* and *tim* closely matched across the entire time course with no significant difference at any time point (Figure 1, Table 3). Both transcripts exhibited a major increase in association with the onset of dusk or subjective dusk.

### 3.2. P. xylostella Individual Adult Locomotor Rhythms at Constant Temperature

*P. xylostella* diel locomotor behaviour was assessed on a sugar/agar substrate using Trikinetics LAM25 monitors at a constant 17 °C for D/D and L/L conditions as well as 12:12, 14:10, 16:8, 18:6 and 20:4 L/D cycles and 6:6:6:6 L/D/L/D cycles (see Figure 2, Appendix A, Table 4, Appendix A). While locomotor behaviour was rhythmic under all L/D cycles, this was not the case for constant D/D or L/L conditions. Moreover, diel activity profiles did not reveal anticipatory features. The activity was increased immediately upon transition from light to dark and then dropped off to a lower level, while the onset of light triggered an immediate and sustained drop in activity. A prior study observed that when placed in 6:6:6:6 L/D/L/D photoperiods, wild-type, but not arrhythmic mutant *Drosophila melanogaster*, exhibited preferential rhythmicity in the circadian range rather than the entrained 12 h L/D cycle [61], such that increased locomotor activity was observed every other 6:6 L/D cycle. However, as is evident from the activity profiles and actograms (see Figure 2, Appendix A) as well as comparative analyses of activity levels during the first and second 6:6 L/D phases per cycle (see Appendix A), this was not the case for *P. xylostella*. In fact, the differences between the first and second 6:6 L/D phases were no more pronounced than those for the subjective dark versus subjective light phases of the D/D condition, where behaviour was arrhythmic. Thus, no evidence of circadian control of adult locomotor behaviour was uncovered. Instead of clock-mediated rhythmicity, the observed diel locomotor activity patterns exhibited two other regulatory features: (1) light-mediated repression and (2) homeostasis. Beyond the qualitative changes in the activity profiles and actograms, light-mediated repression was demonstrated quantitatively in the significantly reduced activity counts under L/L conditions (Figure 2B, Appendix A) and the preferential activity during the dark phase of L/D cycles versus the subjective dark phases of D/D or L/L cycles (Figure 2D, Appendix A). The data also indicated homeostatic control of diel locomotor activity in the dark as total diel activity was relatively similar across L/D cycles with dark phases varying from 4 to 12 h (Figure 2B, Appendix A), while the dark-phase activity rate was lowest in D/D (Figure 2C, Appendix A). Given very strong light-mediated repression, diel dark-phase activity would be expected to increase proportionally with dark-phase length in the absence of homeostasis. In contrast, homeostatic maintenance of a fixed diel activity level would require proportional decreases in dark-phase activity rates with increasing diel dark-phase length. In fact, the observed adult male and female locomotor activity data were best fit by an intermediate model exhibiting some level of homeostasis (Figure 3). Though light repression was strong, it was incomplete as the light phase contributed more to total activity in L/D cycles with short dark phases (18:6, 20:4) versus those with equal amounts of light and dark (12:12, 6:6:6:6; Figure 2D, Appendix A).

To follow up on observations by Wang and colleagues [49], who reported weak but significant circadian rhythmicity for the locomotor behaviour of adult male *P*. *xylostella* of the Geneva 88 strain under 20 °C D/D conditions following prior L/D entrainment, we repeated D/D locomotor analysis for adult male *P. xylostella* of the ROTH strain at 20 °C following L/D entrainment at the same temperature. Again, no behavioural circadian rhythmicity was detected (87.5% arrhythmicity; average RRP 0.82 ± 0.038; Appendix A).

### 3.3. Diel Locomotor Activity Patterns of Individual Adult P. xylostella Males Under Simulated Field Conditions

To better assess diel *P. xylostella* locomotor behaviour under more natural combined light and temperature cycles, we recreated environmental cycles mimicking UK semi-field conditions recorded in Kent in April and June (see [50]). Diel locomotor activity patterns recorded for individual adult males exhibited apparent rhythmicity under both conditions, with both light- and temperature-associated features. Peak activity occurred soon following dark-phase onset, whereas light-phase onset triggered a reduction in activity (Figure 4, Table 5). A significantly larger proportion of diel activity occurred in the dark phase under simulated April conditions, which is consistent with the longer dark phase under these conditions as observed dark-phase activity rates were similar (Figure 4).

### 3.4. Diel Locomotor Activity Behaviour of Individual P. xylostella Males in Response to Temperature Cycles in Constant Light or Constant Dark Conditions

Given the temperature-associated diel locomotor activity features observed under simulated field conditions, we considered the possibility that temperature cycles might induce diel locomotor rhythms in D/D and/or L/L conditions. Significant diel rhythmicity was observed in D/D conditions in the presence of a gradually ramping 14–22 °C temperature cycle, but this was not the case in L/L (see Figure 5, Table 6). Locomotor activity was strongly suppressed by the presence of constant white light and residual activity did not show diel rhythmicity for the large majority of adult male moths. In contrast, in D/D, increased activity was associated with the 12 h phase surrounding peak temperature as well as the 12 h phase associated with rising temperatures.

## 4. Discussion

### 4.1. Clock Gene Regulation

Our findings that adult *P. xylostella* exhibit rhythmic diel and circadian cycling of *per* and *tim* matches observations of other Lepidopterans [62,63,64,65]. Nocturnal moths representing a variety of Lepidopteran families were previously shown to exhibit diel and circadian *per* and *tim* mRNA rhythms with peaks during (subjective) night. This includes the cotton bollworm *Helicoverpa armigera* [65], Mediterranean flour moth *Ephestia kuehniella* [63], Chinese oak silk moth *Antheraea pernyi* [66] and silkworm *Bombyx mori* [67]. We can now add *P. xylostella* as a representative of the superfamily Yponomeutoidea to this list. Moreover, *per* and *tim* transcripts of the day-active monarch butterfly *Danaus plexippus* have also been found to exhibit similar rhythms in L/D and D/D [33,62], confirming that the circadian control of these transcripts is conserved and can be associated with Lepidoptera occupying different temporal niches [68,69]. Under D/D conditions, both *per* and *tim* had lower amplitudes in adult male *P. xylostella* relative expression values compared to L/D, with a significantly reduced rhythmic amplitude for *per* expression. Such damping of clock gene expression in D/D versus L/D conditions has been observed in other Lepidoptera such as *Ephestia kuehniella* [63] and may reflect both a light-mediated enhancement of rhythms as well as a loss of synchrony in peripheral clocks. Tissue-specific studies of molecular circadian rhythms under prolonged free-running conditions may clarify this.

### 4.2. Absence of Circadian Rhythms in P. xylostella Adult Locomotor Behaviour

Various aspects of Lepidopteran behaviour are known to exhibit control by the circadian clock (as reviewed by [70]). Notably, circadian regulation of adult locomotion or flight has been reported for *Ephestia kuehnellia, Antheraea pernyi, Hyalophoroa cecropia, Samia cynthia ricini*, *Manduca sexta* and *Hyles lineata* [63,71,72]. Though many studied Lepidoptera express rhythms in D/D, there are some that also show a notable weakening of rhythms, including *Hyles lineata* [71] and female *P. interpunctella* [73]. We failed to observe circadian control of adult behaviour in *P. xylostella*. Under constant conditions, adult moths failed to exhibit detectable circadian rhythmicity in their locomotor behaviour. This was true for both males and females at 17 °C in D/D or L/L, as well as males at 20 °C in D/D. Moreover, there was no evidence of circadian modulation of diel behaviour, which is evidenced by anticipatory increases in activity ahead of light/dark transitions [74] or circadian modulation of non-24 h L/D cycles [61]. In fact, the complete absence of circadian rhythmicity also contrasts with a prior report of weak but significant circadian locomotor rhythms in adult male *P. xylostella* [49]. It should be noted that a number of experimental differences may account for this discrepancy as our work differed in the strain (ROTH versus Geneva 88) and culture method used (culture on *Brassica rapa* versus artificial diet), and there were also subtle differences in the adult setup for locomotor behaviour.

### 4.3. Light-Mediated Repression of P. xylostella Moth Locomotion

Adaptations to avoid activity under light conditions represent a widely adopted survival strategy [75,76]. Avoiding these activities during light phases is common due to increased risks of predation under light exposure [77]. Moreover, key physiological processes in moths are known to be disrupted by artificial light [78,79,80]. Light avoidance is particularly well known among nocturnal moths [69] and it has been noted in the study of other behaviours in *P. xylostella* with over 70% of mating and oviposition taking place within the first half of the dark phase in 16/8 conditions [81]. Consistent with these broader principles, our adult moth locomotor activity records demonstrated strong non-circadian responses to changes in lighting conditions with activity levels showing a sudden and sustained decrease upon exposure to light and an immediate temporary increase upon the onset of darkness. Our behavioural analyses under simulated field conditions with dimmable lighting are compatible with *P. xylostella* exhibiting a high level of light sensitivity, with strong activity responses associated with subtle changes in light exposure at the beginning of dawn and the end of dusk. *P. xylostella* may preferentially avoid activity at dawn as predators may use morning light to hunt lingering nocturnal prey, with previous research showing higher predator foraging efficiency around this time [82].

### 4.4. Homeostatic Regulation of P. xylostella Locomotor Activity

Sleep is known to exhibit a homeostatic component that is independent of circadian control, where the possible accumulation of ‘sleep debt’ leads to organisms attempting to maintain a certain level of sleep on average [83,84]. Our analyses indicated that homeostasis of diel locomotor activity levels in adult *P. xylostella* at 17 °C across a range of photoperiods with only L/L conditions exhibits a dramatic reduction in activity levels. This phenomenon may at least be in part accounted for by the tendency of adult *P. xylostella* to exhibit a temporary peak in locomotor activity shortly after the onset of darkness.

### 4.5. Temperature Regulation of diel P. xylostella Adult Locomotor Activity

Our results showed temperature-driven control of adult *P. xylostella* locomotor activity. This may be relevant to impact of both diel and seasonal environmental temperature rhythms. In addition, the impact of temperature on *P. xylostella* activity is relevant in relation to migration across temperature zones [85,86] and the impact of climate change [3]. Notably, the impact of temperature on locomotor activity was abrogated in L/L conditions but appeared to persist in more complex environmental L/D cycles simulating field conditions. This observation could indicate that prolonged light exposure is required for this effect. In many animals prolonged exposure to L/L disrupts the circadian clock and associated behavioural rhythms. In insects such as the monarch butterfly *D. plexippus* [41] and the fruit fly *D. melanogaster* [87], this molecular and behavioural arrhythmicity is mediated via the circadian photoreceptor dCRY. dCRY has an orthologue in *P. xylostella*, which was recently genetically disrupted [88]. It would be of interest to determine whether *dcry* mutant *P. xylostella* exhibits temperature-mediated modulation of locomotor activity in L/L.

### 4.6. Disassociation of Locomotor Behaviour Rhythms and Circadian Clock Function

It is unclear why *P. xylostella*’s locomotor activity is governed by light-mediated repression, temperature modulation and homeostatic regulation without exhibiting circadian rhythmicity. One possibility is that the ROTH strain used in this study lost some of its original circadian behaviour after decades of culture under laboratory conditions. Nevertheless, independent observations for the Geneva 88 strain also showed either arrhythmic or weakly circadian behaviour suggesting that *P. xylostella* locomotor behaviour may be largely controlled by alternative means [49]. Perhaps, a homeostatic mechanism in combination with responses to environmental cues is sufficient to drive the onset of temporary peaks in locomotor activity at the onset of darkness. Beyond abiotic time cues such as light and temperature, *P. xylostella* may also respond to rhythmic signals emitted by its host plants. Studies have repeatedly indicated a close relationship between the circadian rhythms of Lepidoptera and host plants [89,90]. Therefore, further research on how *P. xylostella* interacts with its host plants is needed. In this context, it is of interest that *P. xylostella* sexual activity was altered in response to host plant volatiles [81].

It is possible that locomotor activity is an exception and that other relevant rhythmic adult behaviours including feeding, mating and oviposition are circadian clock-controlled [81,91]. Whether migration, which is photoperiodic in a number of insects [92], is controlled by the circadian clock in *P. xylostella* remains to be determined. As noted above, Campos and colleagues found that the modulating photoperiod had no discernible effect on *P. xylostella* morphological and life history traits in a neotropical population [48]. It is possible that temperature rather than photoperiod acts as a migration cue due to its strong effect on the *P. xylostella* life cycle. In *D. plexippus*, a widely researched migratory lepidopteran, migration is triggered via cooling temperatures and is then maintained using circadian clock-assisted compass mechanisms [41,93]. Insensitivity to photoperiodic entrainments may be advantageous to *P. xylostella*, opting for a direct response to light conditions, due to the large changes in light conditions associated with the scale and speed of *P. xylostella* migration [8,94]. Along with temperature, age and affected nutritional quality of host plants may act as a migration cue as it has been shown that older host plants produced *P. xylostella* adults with favourable migratory traits [95].

## 5. Conclusions

Circadian control of activity rhythms has been described at the behavioural and gene expression levels in both moths and butterflies; however, there can be noticeable species-specific differences between them [62,63,96]. It is therefore unclear how daily timekeeping acts in individual species. We confirmed that light-entrained circadian rhythms of the core clock transcripts *per* and *tim* matched the behaviour of orthologous genes across Lepidoptera [62,63,64,65] and other animals [97,98]. This work could be further expanded by determining the circadian transcriptome of *P. xylostella* to further assess the relatedness of its clock mechanisms to those found in other animals. Our findings now provide evidence for the uncoupling of molecular circadian rhythms from adult locomotor behaviour in the economically impactful pest *P. xylostella*. Instead, light-mediated repression, temperature-modulation and homeostasis were identified as important determinants of adult *P. xylostella* diel locomotor behaviour. Our use of simulated field conditions provides new hypotheses regarding the field activity of adult *P. xylostella*. Follow-up studies may link our results to temporal aspects of integrated pest management strategies for *P. xylostella*, such as those involving chemical treatments and trapping [81,99,100,101,102].

## Figures and Tables

**Figure 1 insects-16-00182-f001:**
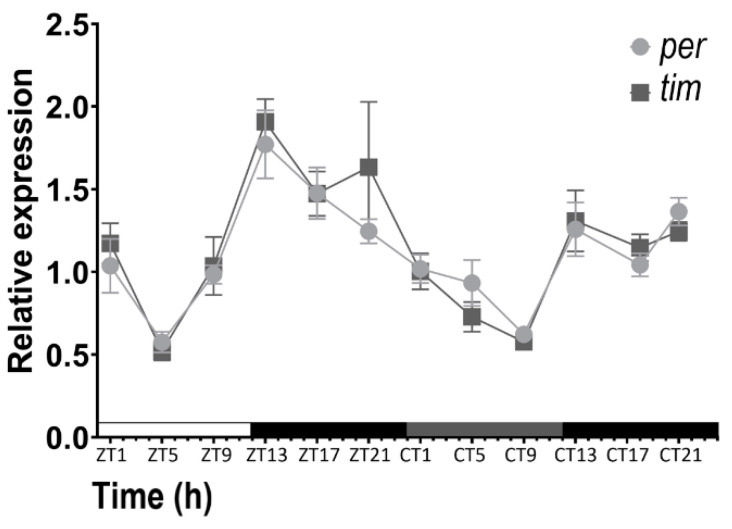
qPCR analysis of *per* and *tim* transcript levels in adult male *P. xylostella* across a 48 h L/D-D/D time course at the indicated L/D (ZT) and D/D (CT) time points. The line graphs show average ± SEM. Relative expression normalised to a reference transcript (*Ef1α*). Six independent biological replicates were used.

**Figure 2 insects-16-00182-f002:**
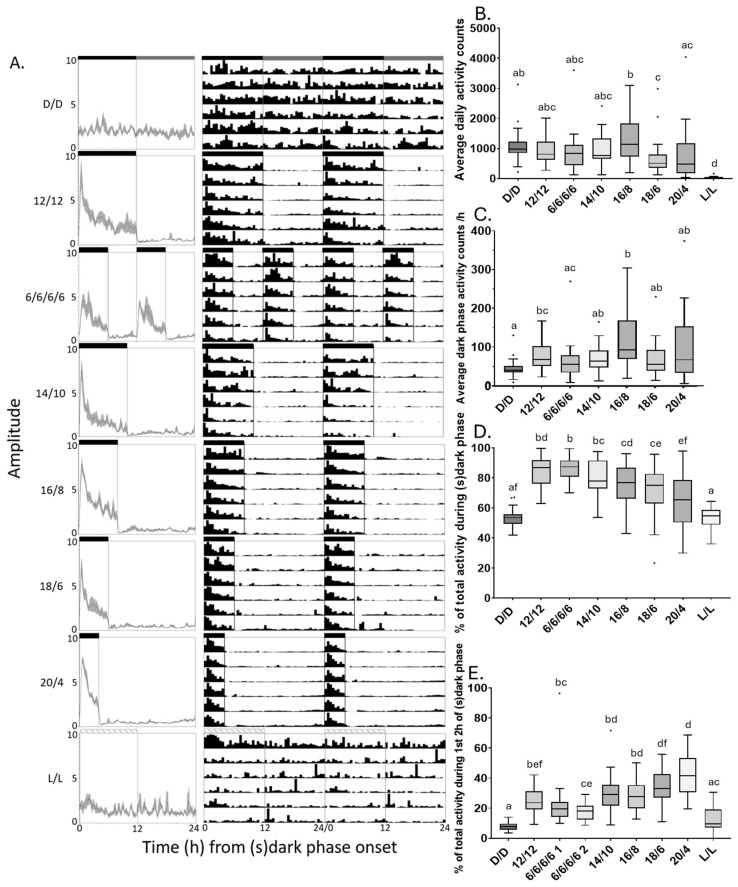
Locomotor behavioural analysis of adult male *P. xylostella* under different lighting conditions. (**A**) Left: normalised average ± SEM daily activity profiles over 6-day intervals for *P. xylostella* adult males. Time is plotted along x-axis in hours (h), starting at the onset of dark or subjective dark phase with normalised activity indicated by the y-axis. Right: double-plotted actograms of normalised average activity of moths over 6 days in 30 min bins. The horizontal bars along the top of individual activity profiles and actograms indicate dark phase (black), subjective dark phase (grey cross hatching), light phase (white) and subjective light phase (grey). (**B**) Total average daily activities, (**C**) average hourly dark-phase activity, (**D**) average % of total activity occurring during (s) dark phase, (**E**) average % of total activity occurring during first 2 h of (s) dark phase. Data points outside Tukey’s range (1.5XIQR (Interquartile range)) are shown. Kruskal–Wallis and Dunn’s multiple comparison post hoc test results show significant differences from each respective data set through letter grouping system.

**Figure 3 insects-16-00182-f003:**
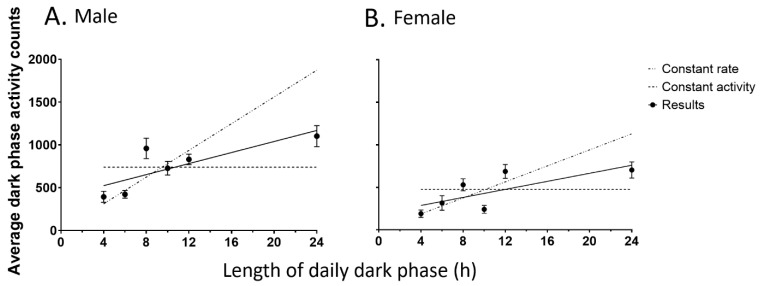
Average dark phase activity as a function of dark-phase duration for *P. xylostella* adult male and female diel locomotor activity. The average ± SEM diel dark-phase activity data are plotted for L/D cycles with 4, 6, 8, 10 and 12 h of dark as well as D/D cycles. The 12 h data represent a combination of the 12:12 L/D and 6:6:6:6 L/D/L/D combinations. A linear regression fit is shown along with theoretical alternatives assuming near-complete light repression representing either a constant dark-phase activity rate or a constant diel activity level.

**Figure 4 insects-16-00182-f004:**
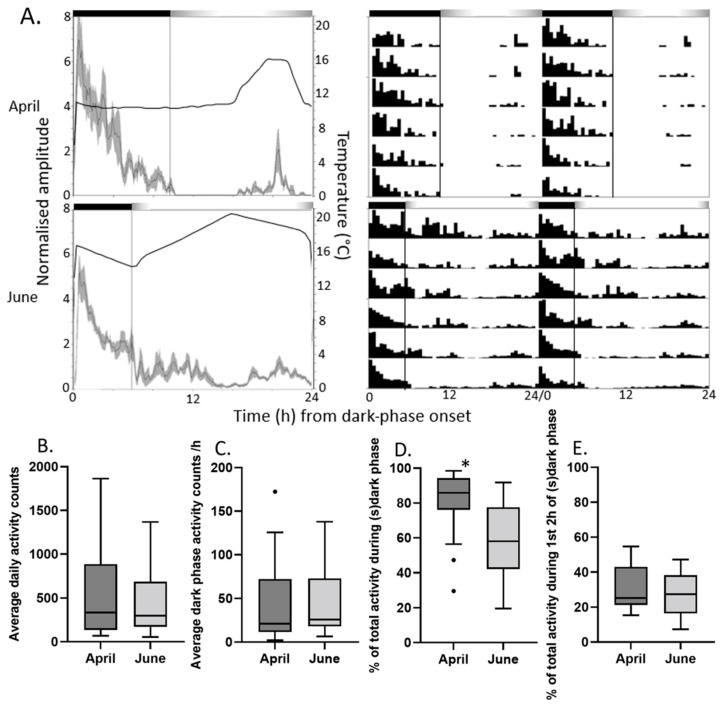
Male adult *P. xylostella* locomotor activity under UK April and June simulation conditions in the lab. (**A**) Left: Average activity profiles show average activity ± SEM (shading) over 24 h of 6-day average activity of 20 *P. xylostella* adult males. Time is plotted along x-axis in hours (h) starting at onset of complete dark phase, with normalised amplitude along y-axis. The black horizontal bar along the top of individual graphs indicates the dark phase with grey-white-grey shaded bars showing the light phase including transitions. A separate line represents the temperature cycle according to the right-hand y-axis. Right: Double-plotted actograms show average activity of moths over 6 days (normalised per 48 h segment) with a 30 min resolution. The environmental light and temperature profiles recorded in the incubators are shown in Appendix A. (**B**) Total average daily counts, (**C**) average hourly dark phase activity, (**D**) average % of total activity occurring during (s) dark phase, (**E**) average % of total activity occurring during first 2 h of (s) dark phase. Data points outside Tukey’s range (1.5XIQR (Interquartile range)) are shown. Mann–Whitney U test fresults show significant differences from respective data set through *.

**Figure 5 insects-16-00182-f005:**
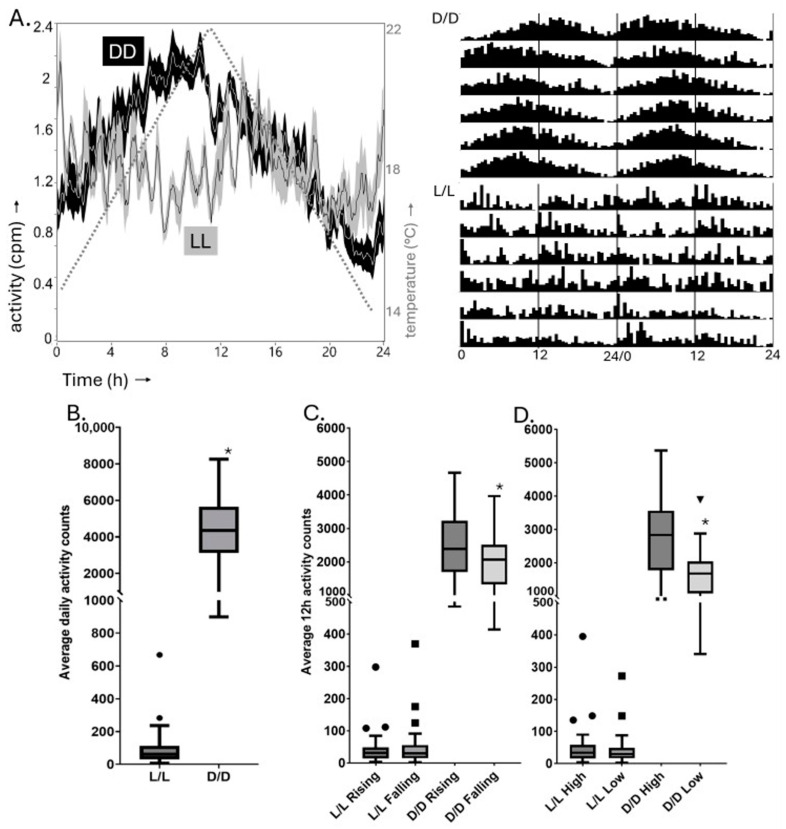
Adult male *P. xylostella* diel locomotor rhythms in D/D or L/L temperature cycle conditions. (**A**) Left: normalised average ± SEM daily activity profiles over 6-day intervals for *P. xylostella* adult males in D/D (black shading) and L/L (grey shading) conditions. Time is plotted along x-axis in hours (h), starting ~1 h following the temperature cycle trough, with normalised amplitude along y-axis. A separate line represents the temperature cycle according to the right-hand y-axis. Right: double-plotted actograms of normalised average activity of moths over 6 days at a 30 min resolution. (**B**) Total average daily activities, (**C**) average activities during the 12 h intervals of rising and falling temperature, respectively, (**D**) average activities during the 12 h non-overlapping intervals surrounding the temperature high and low points. The Y axes are interrupted to allow inspection of the L/L data at higher resolution. Data points outside Tukey’s range (1.5XIQR (Interquartile range)) are shown. Significant differences (Mann–Whitney U test * *p* < 0.05) between conditions in B and between 12 h intervals of the same cycle in (**C**,**D**) are indicated.

**Table 1 insects-16-00182-t001:** Diel environmental conditions for locomotor activity assays. Dawn refers to the phase of increasing light intensity, dusk refers to phase of decreasing light intensity.

Condition	Light Cycle	Temperature
D/D	Constant Dark	17 °C
D/D	Constant Dark	20 °C
12/12	12 h Light, 12 h Dark	17 °C
6/6/6/6	6 h Light, 6 h Dark, 6 h Light, 6 h Dark	17 °C
14/10	14 h Light, 10 h Dark	17 °C
16/8	16 h Light, 8 h Dark	17 °C
18/6	18 h Light, 6 h Dark	17 °C
20/4	20 h Light, 4 h Dark	17 °C
L/L	Constant Light	17 °C
‘April’	4 h Light, 5 h dusk, 10 h Dark, 5 h Dawn	10 °C–16 °C
‘June’	15.5 h Light, 1.5 h dusk, 5.5 h Dark, 1.5 h dawn	14 °C–20 °C
D/D temp	Constant Dark	14 °C–22 °C
L/L temp	Constant Light	14 °C–22 °C

**Table 2 insects-16-00182-t002:** Primer pair sequences and amplicon lengths for *P. xylostella* genes of interest used in qPCR experiments, adapted from [58] and tested using qPCR serial dilution for efficiencies and 5% agarose gel electrophoresis to check expected amplicon lengths.

Primer Pair	Forward Sequence	Reverse Sequence	Amplicon Length (bp)	Efficiencies
*Ef1α*	5′-GCCTCCCTACAGCGAATC-3′	5′-CCTTGAACCAGGGCATCT-3′	162	98.7%
*per*	5′-CCGCGAAAGAACGTCTAAGG-3′	5′-GTGCTCGTGGTCGTGGTTA-3′	118	108.7%
*tim*	5′-ACGCTGCTGAGAAATGGACA-3′	5′-CCGCTATCAGGTCCGATGAC-3′	87	105.9%

**Table 3 insects-16-00182-t003:** CircaCompare analysis of *per* and *tim* qPCR L/D-D/D transcript profiles. The Rhythm p value corresponds to the data’s cosine fit; mesor represents a rhythm-adjusted mean; amplitude reflects the peak-to-trough difference in the fitted curve; phase peak refers to the daily maximum of the fitted curve.

Condition	Rhythm *p* Value	Mesor	Amplitude	Phase Peak
*per*	3.19^−7^	1.11	0.34	16.86
*tim*	3.89^−7^	1.14	0.42	17.12

**Table 4 insects-16-00182-t004:** Periodogram analysis for adult male *P. xylostella* locomotor rhythms. Chi-square periodogram analysis of 6-day data sets for the indicated number (n) of individual males under the indicated conditions. Relative rhythmic power (RRP) was calculated as the ratio between chi-square periodogram amplitude and significance threshold. Condition was found to significantly impact RRP (Kruskal–Wallis test) and Dunn’s multiple comparison post hoc test results are annotated in superscript. # indicates that there was only a single rhythmic individual.

Male Condition (n)	% Rhythmic	Period Length (h)	RRP
D/D (23)	4.3	32.5 ^#^	0.85 ±0.02 ^a^
12/12 (33)	84.8	23.9 ±0.1	1.31 ±0.05 ^bc^
6/6/6/6 (35)	85.7	24.1 ±0.3	1.41 ±0.09 ^bc^
14/10 (20)	75.0	23.9 ±0.1	1.22 ±0.07 ^b^
16/8 (27)	85.2	24.0 ±0.1	1.41 ±0.09 ^bc^
18/6 (33)	87.9	23.8 ±0.2	1.40 ±0.07 ^bc^
20/4 (28)	96.4	24.0 ±0.1	1.72 ±0.07 ^c^
L/L (26)	11.5	27.2 ±2.1	0.86 ±0.03 ^a^

**Table 5 insects-16-00182-t005:** Periodogram analysis for *P. xylostella* locomotor rhythms under simulated UK April and June conditions. Chi-square periodogram analysis of 6-day data sets for the indicated number (n) of individual males under the indicated conditions.

Condition (n)	% Rhythmic	Period Length (h)	RRP
April (20)	90%	24.1 ±0.1	1.68 ±0.09
June (20)	80%	24.1 ±0.1	1.56 ±0.13

**Table 6 insects-16-00182-t006:** Constant dark and constant light temperature cycle data. Chi-square periodogram analysis of 6-day data sets for the indicated number (n) of individual males under the indicated conditions. Significant differences in RRP (Mann–Whitney test *p* < 0.05) are annotated (^a^ versus ^b^).

Condition (n)	% Rhythmic	Period Length (h)	RRP
14–22 D/D (32)	75	24.08 ±0.089	1.36 ±0.090 ^a^
14–22 L/L (29)	17	25.80 ±1.98	0.92 ±0.021 ^b^

## Data Availability

All data are available in this paper.

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
