# Peer review of "Adult Diel Locomotor Behaviour in the Agricultural Pest Plutella xylostella Reflects Temperature-Driven and Light-Repressed Regulation Rather than Coupling to Circadian Clock Gene Rhythms"

_insects, 2025, doi:10.3390/insects16020182_

Round 1
Reviewer 1 Report
Comments and Suggestions for Authors
This is a study that provides an initial assessment of circadian phenotypes in a major lepidopteran pest species. I have to say that the results are not very exciting. The surprise is that unlike almost all insects, under DD animals are not rhythmic. They appear rhythmic under LD conditions but this is simply light induced. The key question is whether the locomotor behavior in DD reflects the underlying molecular cycles in DD of per and tim. The authors interpret their results as a decoupling of the molecular clock with the locomotor phenotype. Unfortunately, the authors cannot state one way or another because they did not perform the qPCR in the second DD cycle. All they show is the first DD (‘cheat’) cycle. They also used whole bodies of male moths for qPCR which might have obscured a stronger cycle in the head as peripheral organs might cycle in different phases.
Specific comments
Given the lack of rhythmic locomotor behavior under DD in Fig 2 one wonders whether the qPCR results in Fig 1 are really rhythmic in DD as implied. The DD extractions were done during the ‘cheat’ cycle, which is the first day of DD. The analysis should have been done in the second day of DD to see whether the transcript rhythms were really rhythmic. The circacompare analysis in Table 3 does not state the period. Also, it’s done on the LD-DD data. Should be done on DD day 2 data. At least a note of caution should be added to the interpretation of the DD results in Fig 1 and Table 3
L240-254 – activity analysis. Can this be written more clearly. Does not reveal much to me. Also Figures relate to males only. Females lay the eggs and are equally important. Table 4 is not very interesting – of course the period is around 24 h in LD cycles. The RRP is not informative except for DD where the animals are arrhythmic. This table should go to supplementaries
Figs 3 and 4 are not very interesting and could also go to supplementaries.
Fig 5 Table 5 simulated natural conditions. Again not much to report….more activity at night in April with longer nights – not surprising
Fig 6 Table 6 shows that under DD the animals can be entrained to temperature but not under LL
To conclude, the authors main contention – decoupling, is not proven. It could be stronger if they used only male heads for qPCR and analyzed the second DD cycle.
Author Response
Thank you very much for taking the time to review this manuscript. We appreciate the points that were raised and have engaged with them to improve the manuscript. Please find the detailed responses below and the corresponding revisions/corrections highlighted/in track changes in the re-submitted files. In consultation with the assistant editor the manuscript was also revised to improve a few isolated language and layout issues.
General Comment 1: The surprise is that unlike almost all insects, under DD animals are not rhythmic.
Response 1: We would make the point that we only make this argument for adult locomotor behaviour as we hold open the possibility that other behaviours and physiological functions will turn out to be clock-controlled in P. xylostella.
General Comment 2: The key question is whether the locomotor behavior in DD reflects the underlying molecular cycles in DD of per and tim. The authors interpret their results as a decoupling of the molecular clock with the locomotor phenotype. Unfortunately, the authors cannot state one way or another because they did not perform the qPCR in the second DD cycle. All they show is the first DD (‘cheat’) cycle. They also used whole bodies of male moths for qPCR which might have obscured a stronger cycle in the head as peripheral organs might cycle in different phases.
Specific Comment 3: Given the lack of rhythmic locomotor behavior under DD in Fig 2 one wonders whether the qPCR results in Fig 1 are really rhythmic in DD as implied. The DD extractions were done during the ‘cheat’ cycle, which is the first day of DD. The analysis should have been done in the second day of DD to see whether the transcript rhythms were really rhythmic. The circacompare analysis in Table 3 does not state the period. Also, it’s done on the LD-DD data. Should be done on DD day 2 data. At least a note of caution should be added to the interpretation of the DD results in Fig 1 and Table 3
Response 2 & 3: The reviewer has concerns regarding the way that the real-time quantitative PCR analysis was conducted and interpreted and suggests that it would have been better to use adult heads on a later day in DD conditions. First, we do provide several analyses of our LD-DD qPCR time course data for per and tim in Table 3, Supplemental Figure S1 and Supplemental Table S1). The separate analysis of the DD day data requested by the reviewer is included in the latter two based on cosine fitting (Supplemental Figure S1) and ANOVA (supplemental Table S1). We interpret the use of the term ‘cheat cycle’ to refer to the reviewer’s concern regarding data from the first day of free-running conditions, which in some organisms can feature known ‘after effects’ from the prior light/dark cycle or the consequences of a ‘transition’ to a new state. While investigating whether molecular rhythms are sustained over longer periods in DD is an interesting question that can help address the damping of oscillators as well as the maintenance of synchrony, we respectfully disagree that it is a requirement to conclude that molecular circadian rhythms are uncoupled from adult locomotor behaviour in P. xylostella. Analysis of prolonged circadian rhythmicity in DD represents an experimental paradigm to assess autonomous circadian rhythmicity, but it does not represent the relevant environment under which circadian clocks normally function or have evolved for most organisms. Instead, circadian clocks act mostly in the context of entrainment to help align and organize internal cycles relative to environmental cycles and each other. Therefore, a persistence of molecular rhythmicity on the first day of DD in the period and phase of the prior LD cycle does point to biologically-relevant circadian clock function. We demonstrate a direct discrepancy between the rhythms of the molecular clock gene transcripts per and tim and locomotor activity patterns during the first day of DD (cf. Fig1 vs 1st DD day behaviour in Figs 2A and Supplemental Figs S2A and S3A). Furthermore, the reviewer acknowledges that in contrast to the strong clock gene oscillations in LD the observed LD locomotor behaviour is ‘simply light induced’, which is supported in our results by the absence of behavioural anticipation or circadian modulation under any of the LD conditions tested. Thus, even in the presence of light entrainment of molecular rhythms there is no evidence of clock-associated locomotor behaviour. In addition, all prior clock gene transcript analyses of molecular rhythmicity in Lepidoptera (as cited in our manuscript) conducted experiments analogous to our own featuring circadian analysis on the first day of DD (Kobelkova et al., 2015 , Yan et al., 2019, Zhu et al., 2008, Merlin et al., 2009). Hence, our selected data format allows our work to be compared to that in other Lepidopteran species. As for our choice to use whole male bodies rather than heads: While it is true that more sustained rhythms could be expected to be found in samples that more selectively contain the circadian pacemaker neurons, it is not certain that whole heads would show a higher level of synchrony. Nevertheless, we observed significant molecular circadian rhythmicity in whole bodies. In conclusion, although the alternative experimental approach proposed by the reviewer would yield potentially interesting new data it is not required to support our conclusion in the context of the presented experiments. We have added new text to acknowledge the benefit of studies with more tissue specificity during conditions of prolonged free run: Tissue-specific studies of molecular circadian rhythms under prolonged free-running conditions may clarify this.' (Lines 354-355).
Specific comment 4: L240-254 – activity analysis. Can this be written more clearly. Does not reveal much to me.
Response 4: Rewritten text: 'The data also indicated homeostatic crontrol of diel locomotor activity in the dark as total diel activity was relatively similar across L/D cycles with dark phases varying from 4 to 12h (Figure 2B, supplemental Figure S2B), while dark phase activity rate was lowest in D/D (Figure 2C, supplemental Figure S2C). Given very strong light-mediated repression, diel dark phase activity would be expected to increase proportionally with dark phase length in the absence of homeostasis. In contrast, homeostatic maintenance of a fixed diel activity level would require proportional decreases in dark phase activity rates with increasing diel dark phase length. In fact, the observed adult male and female locomotor activity data was best fit by an intermediate model exhibiting some level of homeostasis (Figure 3).’ (now Lines 238-247)
Specific Comment 5: Also Figures relate to males only. Females lay the eggs and are equally important.
Response 5: The reason that male data was placed in the main body of manuscript was because some analyses (including the molecular analyses) were only conducted for male moths. As indicated in the methods we preferred extracting males to avoid the consequences of fluctuating egg loads in females (lines 143-144). Separate female behavioural data is found in the supplement (supplemental Figure S2, supplemental Table S2, S3) as well as Figure 3.
Specific Comment 6: Table 4 is not very interesting – of course the period is around 24 h in LD cycles. The RRP is not informative except for DD where the animals are arrhythmic. This table should go to supplementaries
While we agree with the reviewer that 24-periodicity is expected for diel rhythmic behaviour under LD conditions, we would prefer to keep the table in the main body of the manuscript as it quantitatively illustrates the behavioural arrhythmicity under DD and LL conditions with the LD conditions serving as benchmarks. In addition, it is possible to observe weak rhythms with poorly matched period lengths under LD conditions at the individual moth level that are not obvious from an average normalized actogram or activity profile. Hence, there is at least some value in reporting the LD data beyond it serving as positive control.
Specific Comment 7: Figs 3 and 4 are not very interesting and could also go to supplementaries.
Response 7: We have moved Figure 3 to the supplement as our argument regarding the lack of circadian modulation of LD locomotor rhythms is also partially illustrated in Figure 2. We feel that it is helpful, however, to keep figure 4 in place as it helps clarify the point regarding homeostasis, which is less intuitive without sight of the figure.
Reviewer 2 Report
Comments and Suggestions for Authors
The work is very interesting, but raises some questions.
In areas where the seasons change, insect populations have adapted by natural selection to suspend active development in the fall and recover in the spring at certain dates. Individuals that were not adapted to local conditions died of cold and/or starvation. In the second half of summer, the signal of cessation of activity is a decrease in photoperiod, and in spring the temperature must exceed a certain threshold. The pupae of Plutella xylostella overwinter in the soil, and it is important that the soil warms up and that there is nectar for the moth to feed on. Individuals in the population do not develop at the same time, so a certain part of the population dies, while the rest survive temperature fluctuations. In this study, laboratory populations of Plutella xylostella adapted to controlled temperature and photoperiod were used. The response of moths to diurnal variation of light and darkness, may differ from that of natural populations.
The photoperiod is relatively stable for a given latitude and longitude, while the temperature varies greatly in each month, especially in April, when the timing of moth flight is affected by the rate of soil warming and thawing.
In the introduction (partly in the discussion) it is desirable to briefly describe the seasonal cycle of Plutella xylostella, its features depending on latitude, in particular in the region of research (number of generations, dates of the beginning of development after wintering and completion in the fall). In which months is the migration of moths possible? How can their daily activity under controlled conditions be related to migration in nature? Were temperature and lighting in the experiment selected considering long-term data in the study area? Why are the data obtained for Plutella xylostella compared to Drosophila, which has been cultured under controlled conditions for decades?
The last phrase of an Abstract is: ... our analyses show a lack of coupling between the P. xylostella circadian clock and adult locomotor behaviour, which may be relevant in predicting the activity patterns of this agricultural pest. Question: How can the data obtained be used in predicting the activity patterns of this agricultural pest? P. xylostella migrates in the absence of food regardless of whether it migrates during the day or night.
– Keywords “circadian clock” and “locomotor behaviour” are repeated in the Title. Keywords are necessary for search systems and must not double.
– the information given in lines 297-300 must be mentioned in methods, plus latitude, longitude, and more information on P. xylostella seasonal development. There are several generations, when do moths migrate? in each generation? in what month the experiment was done?
References. Most of the references from the list are mentioned in the text.
One source is mentioned twice in the list and the reference in the text is filed differently: citation line 367 a (G. T. Broadhead et al., 2017; line 370 Geoffrey T. Broadhead et al., 2017
Broadhead, G. T., Basu, T., von Arx, M., & Raguso, R. A. (2017). Diel rhythms and sex differences in the locomotor activity of 506 hawkmoths. J Exp Biol, 220(Pt 8), 1472-1480. https://doi.org/10.1242/jeb.143966 507
Broadhead, G. T., Basu, T., von Arx, M., & Raguso, R. A. (2017). Diel rhythms and sex differences in the locomotor activity of 508 hawkmoths. Journal of Experimental Biology, 220(8), 1472-1480. https://doi.org/10.1242/jeb.143966 509
Some titles in brackets are in the format of literature sources. Perhaps they should be described in some way.. (Biosystems, 2019), (BugDorm, 2019), (Clf Plantclimatics, 2019), TriKinetics, 2019a,b) , (ThermoFisher, 2021a b), Primerdesign, 2021...
The references must be placed in the Conclusions (lines 466, 479-480).
Author Response
Thank you very much for taking the time to review this manuscript. We appreciated your interest and detailed comments and have engaged with them to improve our manuscript. Please find the detailed responses below and the corresponding revisions/corrections highlighted/in track changes in the re-submitted files. Note that following consultation with the assistant editor we also included some revisions to improve language and figure layout at this stage.
Comment 1: In areas where the seasons change, insect populations have adapted by natural selection to suspend active development in the fall and recover in the spring at certain dates. Individuals that were not adapted to local conditions died of cold and/or starvation. In the second half of summer, the signal of cessation of activity is a decrease in photoperiod, and in spring the temperature must exceed a certain threshold. The pupae of Plutella xylostella overwinter in the soil, and it is important that the soil warms up and that there is nectar for the moth to feed on. Individuals in the population do not develop at the same time, so a certain part of the population dies, while the rest survive temperature fluctuations. In this study, laboratory populations of Plutella xylostella adapted to controlled temperature and photoperiod were used. The response of moths to diurnal variation of light and darkness, may differ from that of natural populations.
Response 1: We acknowledge that, as with any laboratory-based experiment, there is a chance that natural settings and populations will exhibit differences, which is why we make efforts to include simulations of semi-field conditions. We acknowledge the limitations of using a long-term lab culture strain (though we do maintain it on host plants), but point out that independent behavioural observations with another strain (Wang et al., 2022) are generally consistent with ours (Lines 417-421). A salient point may be that the impact of diel light and temperature on behaviour may interact with one another. This is precisely what we illustrate in Figure 5. We should also point out that the reviewer describes a scenario that would apply to part of P. xylostella’s near global range, but not in populations that reside close to the equator. Those P. xylostella populations that do overwinter under colder conditions would still lack the ability to diapause (Campos et al., 2006; Furlong et al., 2013; Honda, 2003) meaning they would struggle survive harsh winters. In the example of the UK overwintering is not a major source of P. xylostella infestation (climate change obviously risks expanding year round regions).
Comment 2: The photoperiod is relatively stable for a given latitude and longitude, while the temperature varies greatly in each month, especially in April, when the timing of moth flight is affected by the rate of soil warming and thawing.
Response 2: Once again, this all depends on the habitat that P. xylostella finds itself in. P. xylostella in the UK, for example, can experience changes in photoperiod that match both the 12:12 and 18:6 LD conditions used in our studies. Photoperiod is used by other species as signal to adapt (migrate, diapause etc) to potentially changing conditions, but there is some evidence that this would likely not be the case for P. xylostella (Lines 82-84). We do analyze the impact of temperature on locomotor rhythms both separately and in combination with photoperiodic change (Figure 4,5).
Comment 3: In the introduction (partly in the discussion) it is desirable to briefly describe the seasonal cycle of Plutella xylostella, its features depending on latitude, in particular in the region of research (number of generations, dates of the beginning of development after wintering and completion in the fall). In which months is the migration of moths possible? How can their daily activity under controlled conditions be related to migration in nature? Were temperature and lighting in the experiment selected considering long-term data in the study area? Why are the data obtained for Plutella xylostella compared to Drosophila, which has been cultured under controlled conditions for decades?
Response 3: We have reorganised our manuscript to more clearly communicate current knowledge about P. xylostella seasonal migration (Lines 81-87) and its relationship with environmental factors. We have deleted a somewhat out-of-place reference to a Drosophila paper.
Comment 4: The last phrase of an Abstract is: ... our analyses show a lack of coupling between the P. xylostella circadian clock and adult locomotor behaviour, which may be relevant in predicting the activity patterns of this agricultural pest. Question: How can the data obtained be used in predicting the activity patterns of this agricultural pest? P. xylostella migrates in the absence of food regardless of whether it migrates during the day or night.
Response 4: Direct analyses of migratory behaviour fall outside the scope of this paper. Migratory behaviour, responses and regulation would require more specific targeted manipulations (e.g., genetic) and behavioural analyses. Our results regarding the control of diel locomotor activity patterns suggest that P. xylostella may more directly respond to changes in light and temperature or a combination of these rather than rely on a circadian or seasonal timer. If this is confirmed for migratory behaviour, it would improve the ability to predict this behaviour, for example tying it to particular temperature thresholds. While migration may be sustained through both night and day, once initiated, locomotor, mating and oviposition activity upon arrival are still likely to exhibit diel rhythms and this might also be the case for caterpillar feeding. Hence, this work could help improve the ability to predict the onset of migration as well as the timing of P. xylostella behaviours upon invasion of new regions.
Comment 5: – Keywords “circadian clock” and “locomotor behaviour” are repeated in the Title. Keywords are necessary for search systems and must not double.
Response 5: Changed to daily timekeeping and behavioural rhythm (Line 36)
Comment 6: – the information given in lines 297-300 must be mentioned in methods, plus latitude, longitude, and more information on P. xylostella seasonal development. There are several generations, when do moths migrate? in each generation? in what month the experiment was done?
Response 6: A description of the recordings made from semi-field conditions and the incubator equipment used, along with other specific data on the conditions used and setup, is reported in the methods section in 2.2 Locomotor activity assays and Table 1. Recordings were made in Kent, UK (51.2874° N, 0.4400° E) in April and June referencing prior work carried out by Shaw et al. This has now been updated. To be clear, we have replaced the term ‘semi-field’ when referring to the experiments of Figure 5 and Table 6 as these were laboratory simulations using temperature and light profiles recorded under semi-field conditions. As P. xylostella doesn’t go into diapause and is prevalent during these months within their range (peaking populations soon after June) adverse effects from changes in seasonal development shouldn’t be caused by these conditions and therefore have not been reported in the text. As mentioned, P. xylostella development is highly responsive to temperature but the experiments are not capturing developmental data. The P. xylostella used are from the continuous ROTH generations reared in the lab. Further details about migration and it’s affects are outside the scope of this experimentation. It is likelythat migratory generations emerge due to changes in host food signals and abundance and so not every generation of P. xylostella has the capacity to be migratory. Experiments across the paper were carried out across the year but with lab-raised moths under tightly controlled conditions.
Comment 7: One source is mentioned twice in the list and the reference in the text is filed differently: citation line 367 a (G. T. Broadhead et al., 2017; line 370 Geoffrey T. Broadhead et al., 2017
Response 7: Thanks for pointing this out. It is now fixed.
Comment 8: Some titles in brackets are in the format of literature sources. Perhaps they should be described in some way.. (Biosystems, 2019), (BugDorm, 2019), (Clf Plantclimatics, 2019), TriKinetics, 2019a,b) , (ThermoFisher, 2021a b), Primerdesign, 2021...
Response 8: This has been reformatted.
Comment 9: The references must be placed in the Conclusions (lines 466, 479-480).
Response 9: References were added (Lines 450, 460)
Round 2
Reviewer 1 Report
Comments and Suggestions for Authors
I thank the authors for responding to my comments. While I do not find their response to the DD PCR problem particularly compelling, I appreciate that they are following protocols that have been applied to other species that allow direct comparison. I do not want the authors to do additional experimental work as that would be unreasonable. The ms is now improved and I have no further comments to add
Reviewer 2 Report
Comments and Suggestions for Authors
I wrote in the first review "The references must be placed in the Conclusions". It was my technical error. I wanted to say "The references must be placed in the Conclusions". In this case, I leave it to the editor's discretion.